ecology/environmental science

extinction risk, threat pattern, IUCN, conservation, island endemism, birds

**Author for correspondence:**
Lucile Lévêque
e-mail: lucile.leveque@utas.edu.au

# Characterizing the spatio-temporal threats, conservation hotspots and conservation gaps for the most extinction-prone bird family (Aves: Rallidae)

Lucile Lévêque[1], Jessie C. Buettel[1,2], Scott Carver[1] and Barry W. Brook[1,2]

[1]School of Natural Sciences, University of Tasmania, Private Bag 55, Hobart, Tasmania 7001, Australia
[2]ARC Centre of Excellence for Australian Biodiversity and Heritage (CABAH), Australia

LL, 0000-0003-4141-7294; JCB, 0000-0001-6737-7468; SC, 0000-0002-3579-7588; BWB, 0000-0002-2491-1517

With thousands of vertebrate species now threatened with extinction, there is an urgent need to understand and mitigate the causes of wildlife collapse. Rails (Aves: Rallidae), being the most extinction-prone bird family globally, and with one-third of extant rail species now threatened or near threatened, are an emphatic case in point. Here, we undertook a global synthesis of the temporal and spatial threat patterns for Rallidae and determined conservation priorities and gaps. We found two key pathways in the threat pattern for rails. One follows the same trajectory as extinct rails, where island endemic and flightless rails are most threatened, mainly due to invasive predators. The second, created by the diversification of anthropogenic activities, involves continental rails, threatened mainly by agriculture, natural system modifications, and residential and commercial development. Indonesia, the USA, the United Kingdom, New Zealand and Cuba were the priority countries identified by our framework incorporating species' uniqueness and the level of endangerment, but also among the countries that lack conservation actions the most. Future efforts should predominantly target improvements in ecosystem protection and management, as well as ongoing research and monitoring. Forecasting the impacts of climate change on island endemic rails will be particularly valuable to protect rails.

# 1. Introduction

Up to a million species are now threatened with extinction, with many more predicted within the next decade [1]. Along with the unprecedented decline in avifauna abundance, including in once common species (e.g. in North America [2]), the need to understand and mitigate the causes of these biotic contractions is urgent if we wish to avert a major loss of species and ecosystem integrity this century. As the vulnerability to different threats varies across taxa (e.g. between different bird families [3]), there is value in taxon-specific analyses to inform conservation policy and decision making. This has been done for several bird groups including ducks, geese and swans [4], sea birds [5–7] and parrots [8], but gaps exist for many bird taxa, including the rail family.

Rails (Aves: Rallidae) are a widespread bird family, ecologically diverse and inhabiting most habitats worldwide. They are also the most extinction-prone bird family, with 54 to 92% of all species going extinct after their first contact with humans during the mid-Holocene (representing between 200 and 2000 estimated extinct species [9,10]). They went through the second wave of extinction starting in the sixteenth century when European settlers spread worldwide. In this recent extinction wave, they were once again the most represented bird family in the extinct species assemblage (15%, 24 species [11]). Five rails have gone extinct in the last 100 years (*Hypotaenidia pacifica*, *Porphyrio paepae*, *Zapornia palmeri*, *Hypotaenidia wakensis* and *Hypotaenidia poeciloptera*) and the most recent extinction event was recognized in 1994 [12].

As the magnitude, diversity and global reach of anthropogenic activities increase, species can be impacted by an amplified range of pressures and stressors—the International Union for Conservation of Nature (IUCN) Red List now references 12 different types of threats (and 118 sub-categories [12]). As an already extinction-prone bird family, Rallidae could be particularly vulnerable to contemporary threats and continue on their trajectory to extinction. Of the 144 recognized extant species of Rallidae, 13 are near threatened (NT) and 35 are threatened (vulnerable, endangered or critically endangered [12]). To date, our understanding of historical threats and causes of extinctions in rails has been limited, and their pattern of endangerment has not been clearly characterized either spatially (where extinct/threatened species are distributed) or temporally (with respect to changes in threat patterns between extinct and contemporary species). Here, we provide the first such descriptive assessment for Rallidae. We have three main aims: (i) characterize the pattern of threats (historical and modern), focusing on four different spatial scales (globally, on continents, for island endemics and between bioregions); (ii) track the time course of species' conservation status (IUCN status) and gaps in conservation programmes and (iii) identify 'conservation hotspots' for rails by creating a ranking system using rails' conservation values and level of endangerment, so as to identify priority countries and bioregions.

# 2. Methods

## 2.1. Database compilation

We compiled a database for all 144 species of extant rails (including 42 island endemic species and 33 threatened species) and 24 extinct rails using the 2019 version of the IUCN Red List [12] and the 'Guide to the rails, crakes, gallinules and coots of the world' [13]. We used the taxonomic classification followed by the IUCN which included the rallid family of Sarothruridae (some authors consider it separate from the Rallidae family, see [14,15]). The Sarothruridae family contains 15 species, including two threatened species, and is mostly present in the Afrotropics. In preliminary analyses, we evaluated the effect of excluding this group, finding no major effect on results or their interpretations (except a minor effect of Threat rank of countries, see electronic supplementary material, table S1) and, thus, elected to retain the Sarothruridae in the Rallidae family. We considered island endemic species as those restricted to one or a group of adjacent islands. Species considered 'Data deficient' were excluded from the analysis (electronic supplementary material, table S2). Similarly, the New Caledonian rail (*Gallirallus lafresnayanus*) and the Samoan moorhen (*Pareudiastes pacificus*) are two 'Critically endangered' rail species that have not been seen with certainty since the nineteenth century and are suspected to be extinct [12], so they were considered 'extinct' in this study (electronic supplementary material, table S2).

## 2.2. Spatial and temporal patterns of threats

We compiled the contemporary threats for all extant rails and historical causes of extinction for all extinct rails, available from the online IUCN database for each species (http://iucnredlist.org). The IUCN's

threats classification organizes threats in a hierarchy (https://www.iucnredlist.org/resources/threat-classification-scheme).

### 2.2.1. Spatial threats patterns of extinct and threatened rails

First, we compared the historical causes of extinction (extinct rails) with the contemporary threats (threatened rails) to assess temporal changes in the threat pattern and changes in the spatial distribution of threatened and extinct species.

Then, we described the number of threats impacting threatened and extinct species for different spatial scales (globally, on continents, on islands, between bioregions). For this analysis of threat diversity, we used the threats' first sub-category as the level of threat (e.g. a rail being impacted by sub-categories '5.1. Hunting & collecting terrestrial animals' and '5.3. Logging & wood harvesting' within the threat '5. Biological resource use' was considered as threatened by two threats). A total of 45 threats were considered, but only 28 were relevant to rails.

### 2.2.2. Spatial threats patterns of all extant rails

We assessed the spatial pattern of contemporary threats for all extant rails (whether or not they were considered threatened by the IUCN) by descriptively comparing the type of threats and their impact: (i) globally, (ii) on islands, (iii) on continents and (iv) by bioregions (biogeographic realms, hereafter 'bioregions': Australasia, Oceania, Nearctic, Neotropics, Palaearctic, Afrotropics and Indomalaya; defined following Olson *et al.* [16]). Oceania was grouped with the Australasia bioregion because of its few extant species (five, including three present across the two bioregions).

We calculated each threat's impact score using the *Threat Impact Scoring System* (*IUCN – CMP Unified Classification of Direct Threats*, version 3.2) proposed by the IUCN [12] that defines an overall impact based on scope, severity and timing, and ranges within 'Negligible', 'Low', 'Medium' and 'High' impact (https://www.iucnredlist.org/resources/threat-classification-scheme; electronic supplementary material, figure S1). If a threat had a different impact in two sub-categories, we used the higher impact for the classification. We included 'Past' and 'Ongoing' impact to illustrate the temporal evolution of threats ('Future' was not considered).

For this analysis, we used the first category as the level of threat (e.g. 1. 'Residential & commercial development' and 2. 'Agriculture') to be more informative. We split the threat '5. Biological resource use' in two categories: 'Hunting & collecting terrestrial animals' (5.1.) and 'Logging & indirect effects' (regrouping '5.2. Gathering terrestrial plants', '5.3 Logging & wood harvesting' and '5.4. Fishing & harvesting aquatic resources') to provide relevant conservation policy. The sub-category '10. Geological events' was not included because it was not listed for any species.

## 2.3. Conservation status and gaps

To identify gaps in conservation efforts for rails, we summarized the IUCN's *Conservation actions classification scheme* v. 2.0 (https://www.iucnredlist.org/resources/conservation-actions-classification-scheme) and *Research needed classification scheme* v. 2.0 (https://www.iucnredlist.org/resources/research-needed-classification-scheme): (i) globally, (ii) per country and (iii) per bioregion, following Olah *et al.* [8]. Countries and overseas territories were grouped together for country-level analysis (electronic supplementary material, table S3). These schemes specify the conservation actions and research needed for each species and were available on the online IUCN database (http://iucnredlist.org). We gathered the possible classifications in five categories: (i) 'Research & Monitoring', (ii) 'Ecosystem Protection & Management', (iii) 'Species Management', (iv) 'Education & Awareness' and (v) 'Law & Policy' (i.e. legislative protection).

(i) 'Research & Monitoring' category was formed by grouping all categories in the *Research needed classification scheme*: '1. Research', '2. Conservation Planning', '3. Monitoring' and '4. Other'. (ii) 'Ecosystem Protection & Management' category was formed by grouping the first two categories, '1. Land/water protection' and '2. Land/water management', in the *Conservation actions classification scheme*. (iii) 'Species Management', (iv) 'Education & Awareness' and (v) 'Law & Policy' corresponded directly to each remaining category in the *Conservation actions classification scheme*. Furthermore, category '6. Livelihood, economic & other incentives' of the *Conservation actions classification scheme* was not represented as it did not appear in any of the studied species.

To assess whether conservation status had improved or deteriorated over time, we characterized changes in species' IUCN conservation status since 1988 (first IUCN Red List assessment) through to 2019 [12]. Changes in conservation status could happen any year between 1988 and 2019. In cases where status changed more than once, we only used the most recent change. We considered 'Unknown (LR/LC)' as Least Concern and 'Unknown (LR/NT)' as Near threatened when they were the assessments preceding a new status. However, we ignored it when included between two assessments. For example, if a species' conservation status history was 1996: Vulnerable (VU); 2000: Unknown (LR/NT); 2004: Near threatened, we considered that the species improved from Vulnerable to Near threatened.

## 2.4. Rails 'conservation hotspots' and priority rankings

Rallidae combine species with unique evolutionary traits (e.g. flightlessness and endemism) and species with elevated vulnerability (24% of the species are threatened); however, to date, there is no framework that accounts for these important aspects in terms of conservation priority. To identify areas with high conservation interest and/or those that deserve improvements in their protection, we conceived a ranking system to classify both world bioregions and countries of high conservation priority for rails (conservation hotspots). Analysis at the country level grouped countries and their overseas territories together (electronic supplementary material, table S3). We created two categories that we deemed useful to reflect important rail conservation: 'Heritage' and 'Threat'.

The 'Heritage' category aimed to account for species with high conservation value, using endemism and unique evolutionary trait (flightlessness). It was calculated by ranking the number of rail species being either flightless (first), island endemic (second) or country endemic (third). When relevant, species were classified by one of the above attributes, in this hierarchical order. For instance, a flightless rail that is also endemic to an island would only be classified as 'flightless', and an island endemic rail that is also endemic to one country would only be classified as 'island endemic'.

The 'Threat' category was calculated by ranking the number of threatened species (first) and the number of species with a worsened IUCN status since their last change in status (second).

In cases where ranks were equal, they were split using the country/bioregion's richness in rail species as the tie breaker (i.e. higher richness would get the higher rank of conservation importance). In this rank system, rank one represents the highest rank for conservation priority. The rank classifications were not made to adequately differentiate between two entities as close scores would not illustrate a true difference in conservation priority. We recommend identifying entities with the highest conservation priority as the ones present in both top classifications for the two ranks, or as parts of a top five, top 10 or top 20.

# 3. Results

## 3.1. Spatial and temporal pattern of threats

### 3.1.1. Spatial threats patterns of extinct and threatened rails

All recent rail extinctions (post AD 1500) occurred on islands ($n = 26$ species: 24 officially extinct and the New Caledonian rail and Samoan moorhen, considered extinct in this study) and at least 77% were of flightless species (one species was flying and five were not documented sufficiently to be confident about their flight ability [12,13]). Rail extinctions were linked to a total of six different threat types (IUCN sub-categories).

Of all threatened rail species ($n = 33$), the vast majority were island endemics (67%, table 1 and figure 1), of which 59% were flightless (39% of all threatened species). The number of threats for threatened island rails (table 1) was more than twice the known threats that caused extinctions (mean ± s.e.: $1.5 ± 0.1$ threat per extinct species).

Most rail extinctions occurred in the Pacific Ocean (65%, Australasia: 8 species, Oceania: 9 species), with 19% in the Indian Ocean (Mascarene islands, 5 species) and 15% on remote Atlantic islands (4 species). Contemporarily, Australasia/Oceania and the Neotropics were the bioregions that host the highest number of threatened rails, representing 35 and 32 species, accounting for 30 and 21% of their total rail species richness, respectively. Indomalaya and Nearctic were the bioregions with the less threatened rails (one species each, respectively, *Gallirallus calayanensis* and *Laterallus jamaicensis*).

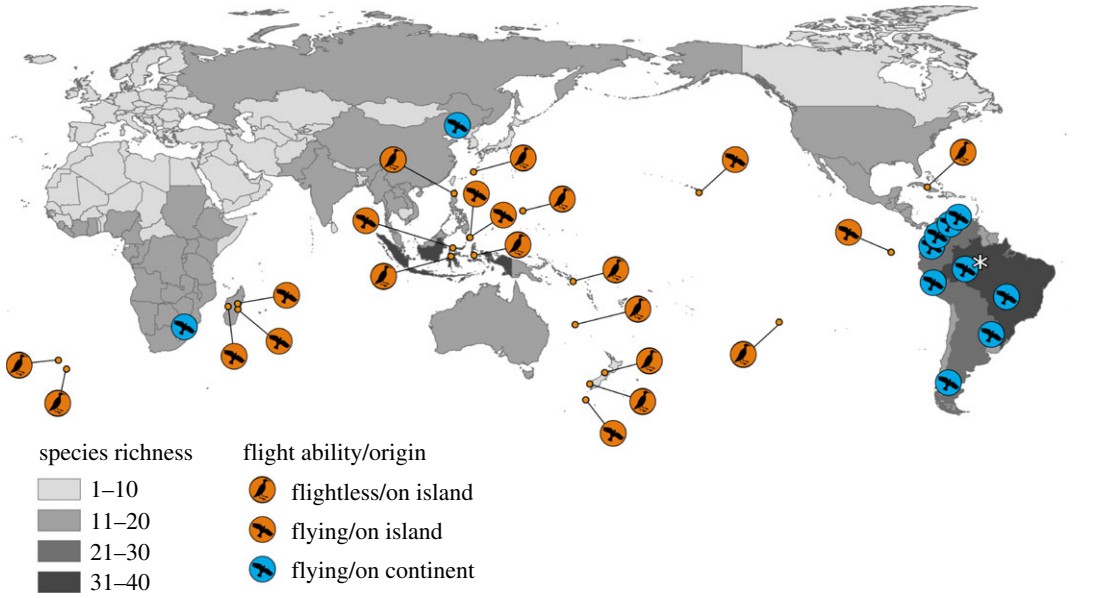

**Figure 1.** Global distribution of threatened rails, highlighting their origin: continental (blue, $n = 11$), island endemic (orange, $n = 22$) and flying ability (flightless: $n = 13$). The data points are placed at the centroid of the species' range, as found on their IUCN profiles. The asterisk symbolizes the only continental species (Black rail, *Laterallus jamaicensis*) whose distribution spans across the American continent, including North, Central and South America. The grey shading represents countries' total rail richness (excluding their overseas territories). Projection information: WGS84, centred on 150°E.

**Table 1.** Number of threats impacting threatened species at different spatial scales.

| scales (% threatened sp.) | median (Q1–Q3) | min.—max. | mean ± s.e. |
|---|---|---|---|
| globally (24%) | 4 (3.0–6.0) | 1–11 | 4.7 ± 0.4 |
| continents (11%) | 4 (4.0–8.0) | 4–11 | 6.2 ± 0.8 |
| islands (52%) | 4 (2.3–5.0) | 1–9 | 4.0 ± 0.5 |

Introduced predators and over-hunting were the main drivers to island rail extinctions: introduced predators were the single cause in 27% of the cases (and involved in combination with other threats for 69% of the extinctions) and over-hunting was the only responsible factor in 23% of them (and involved with other threats for 62% of the extinctions). Habitat loss contributed to 23% of extinctions, as a consequence of 'Natural system modifications' (fire & fire suppression, other ecosystem modifications), 'Agriculture & aquaculture' (livestock farming & ranching, annual & perennial non-timber crops), 'Invasive and other problematic species, genes & diseases' and 'Climate change & severe weather' (storms & flooding).

### 3.1.2. Spatial threats patterns of all extant rails

Contemporarily, agriculture, invasive species, and logging were the three predominant threats to the extant rails globally (impacting 19–26% of species, figure 2a). No threats were found to be of 'High impact' ('Rapid' or 'Very rapid' severity, combined with 'Majority' or 'Whole' scope; figure 2a; electronic supplementary material, table S1). 'Invasive & problematic species', and 'Climate change & severe weather' were the most prevalent threats with a medium impact (figure 2a).

Threats due to agriculture mostly impacted species found in the Afrotropics and the Neotropics, those of invasive species in Australasia/Oceania, and logging impacted mostly species in the Afrotropics and Australasia/Oceania (number of species, figure 2b). Globally, less than 20% of rails are impacted by 'Natural system modifications'; however, it is an important factor in the Palaearctic, where it impacts 60% of that region's rails (figure 2b).

For island rails, threats associated with invasive species, hunting and logging prevailed (38–62% of species), with invasive species (specifically, introduced predators) predominating this group

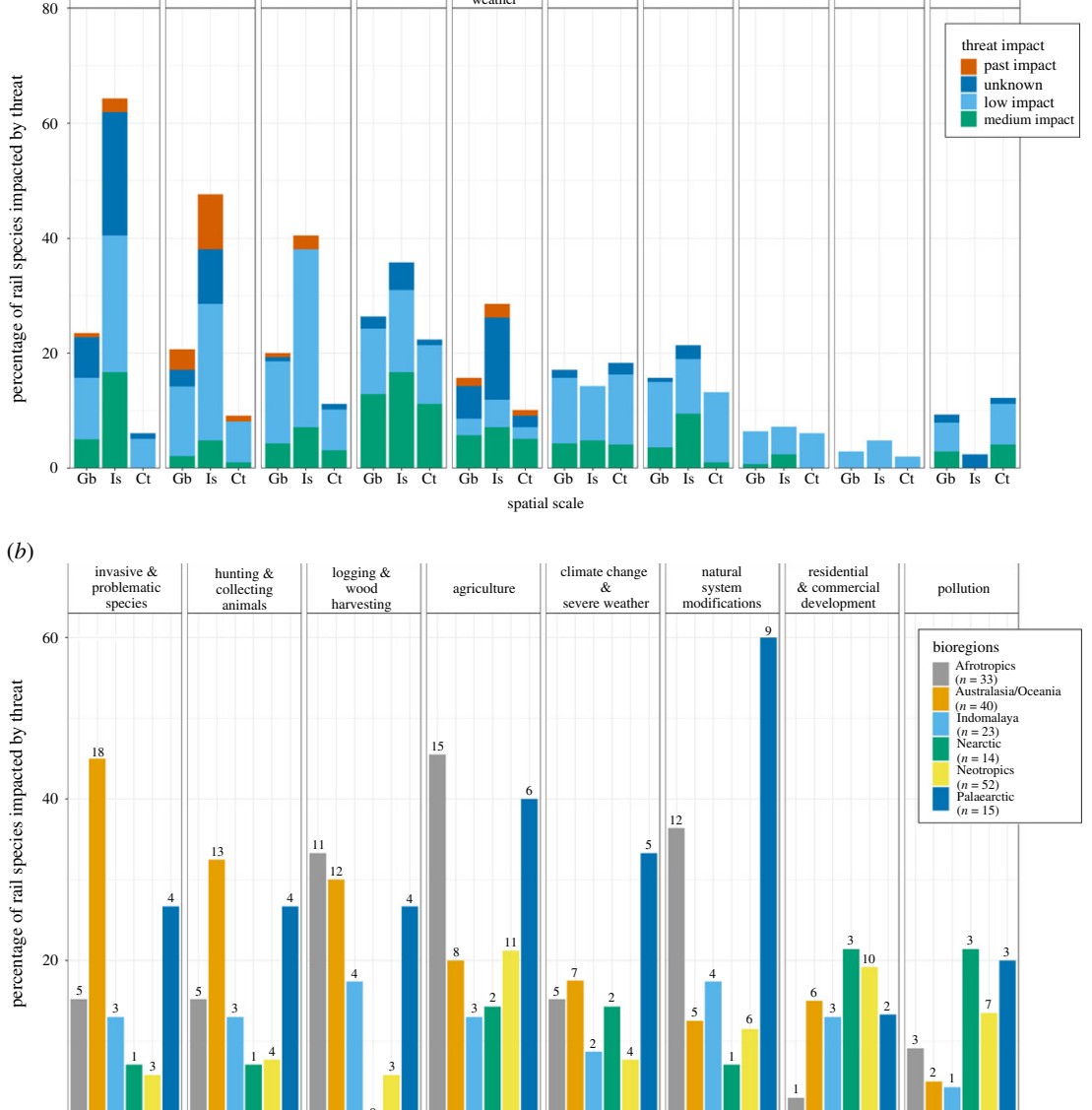

**Figure 2.** (*a*) Threat diversity and intensity, measured as the impact score proposed by the IUCN Red List of Threatened Species, for rail species at a global scale (Gb; *n* = 140), for island endemic rail species (Is; *n* = 42) and for continental rail species (Ct; *n* = 98). Categories 'Other' and 'Energy production & mining' are excluded. (*b*) Threat diversity and percentage of rails impacted for each bioregion. Only threats impacting over 10 species are presented ('Other' and 'Energy production & mining' are excluded). Bar labels represent the number of species per bioregion.

(figure 2*a*). 'Pollution' and 'Natural system modifications' (e.g. fire, dams and water abstraction) were the threats that were more threatening on continents than for island endemics (figure 2*a*), while 'Energy production & mining' and 'Other' were only found in continental species.

## 3.2. Conservation status and gaps

Following the IUCN classification, 'Research & Monitoring' and 'Ecosystem Protection & Management' were the two most important gaps in conservation efforts directed towards rails, both globally (table 2) and at the country scale (electronic supplementary material, table S4). They were especially needed in Australasia/Oceania, Palaearctic and Nearctic, where up to a quarter of the species required such effort (table 2). Non-threatened species accounted for 25–79% of the species that required more efforts in 'Research & Monitoring' and 'Ecosystem Protection & Management' categories. Moreover, each

**Table 2.** Percentage of species for which conservation gaps were identified by the IUCN Red List (2019), by bioregion. Values in brackets give the proportion of threatened rail species in each conservation gap.

| bioregion (total no. of species) | research & monitoring | ecosystem protection & management | species management | education & awareness | law & policy |
|---|---|---|---|---|---|
| Australasia/ Oceania (40) | 40.0 (22.5) | 27.5 (20.0) | 22.5 (12.5) | 15.0 (5.0) | 7.5 (7.5) |
| Palaearctic (15) | 26.7 (13.3) | 20.0 (13.3) | 20.0 (13.3) | 13.3 (6.7) | 13.3 (0.0) |
| Nearctic (14) | 28.6 (7.1) | 28.6 (7.1) | 14.3 (7.1) | 7.1 (0.0) | 0.0 (0.0) |
| Neotropics (52) | 26.9 (21.2) | 23.1 (19.6) | 9.6 (7.7) | 1.9 (1.9) | 0.0 (0.0) |
| Afrotropics (33) | 18.2 (12.1) | 21.2 (12.1) | 6.1 (0.0) | 6.1 (3.0) | 3.0 (0.0) |
| Indomalaya (23) | 13.0 (4.3) | 8.7 (4.3) | 4.3 (4.3) | 4.3 (4.3) | 8.7 (0.0) |
| globally (140) | 31.7 (20.0) | 27.1 (20.0) | 16.4 (10.7) | 9.3 (5.0) | 4.3 (2.1) |

threatened species had, on average, $2.5 \pm 0.2$ (s.d.) conservation actions required, with 45% of them needing more than two types of actions.

In the 31-year period between the first Red List assessment (1988) and the most recent (2019), the IUCN conservation status of 11 species worsened, including six species that got allocated a threatened status for the first time (electronic supplementary material, table S5). Of those 11 species, eight were endemic to islands and three were flightless. Conversely, the conservation status improved for seven species, including two downgraded to 'Least Concern', one species removed from the threatened category and one species downgraded from 'Extinct in the Wild' to 'Critically Endangered' thanks to a managed relocation (electronic supplementary material, table S5).

## 3.3. Rail 'conservation hotspots' and priority rankings

We identified 13 countries present in both the top 20 for Heritage and Threat ranks (table 3). Of them, we could define the five top priority countries (found in both top 10 for Threat and Heritage ranks): Indonesia, New Zealand, the USA, the UK and Cuba. Indonesia, with its overseas territories included, had the highest rail richness globally (23%). All island endemics were found in 14 countries (61% of them were in Australasia/Oceania) and all flightless species in 10 countries (70% of them were in Australasia/Oceania). Of the countries from the top 20 Threat rank, 70% were continental (the USA and Ecuador were the only continental countries to have island endemics, table 3). Australasia/ Oceania was the most important 'conservation hotspot' for rails under our classification system (being top one for 'Heritage' and 'Threat' ranks, electronic supplementary material, table S6).

# 4. Discussion

## 4.1. Threat patterns

While most threatened bird species are found on continents [17], the rails show a different pattern. All extinct and most threatened rail species are island endemics, including most flightless species. Most of the past rail extinctions occurred on the Pacific islands, directly linked to the fact that they were the support for the largest radiation of rails, reaching high levels of endemism [9,18–20]. Flightlessness is an evolutionary trait that has been found to make bird species more extinction-prone during different waves of extinction [21,22], especially rails [9,10]. Moreover, it also appears as an important contributor to the vulnerability of rails to contemporary threatening processes.

From the six threat types recognized from past rail extinctions, anthropogenic activities have diversified to impact rails with up to 11 different threats, as recognized by the IUCN, with some of these threats affecting up to 60% of all island rails. The three key threats that we identified for extant rails—impacting about a fifth of all rails globally—are agriculture, invasive species and logging, a pattern that is consistent with other bird species worldwide [23]. If we also include hunting, these four are also the most impactful threats for island endemic rails (with a different order of impact, and invasive predators

**Table 3.** Classification of countries of highest priority for rail conservation, ordered and ranked by total numbers of: (i) threatened, and (ii) species with a declined conservation status (Threat rank), (iii) flightless, (iv) island endemic, and (v) country endemic species (Heritage rank). *Ex aequo* countries (equal rank) were split using their total number of species (higher priority for higher richness). Lower values in ranks indicate a higher conservation priority, with rank one representing the highest rank. See electronic supplementary material, table S7 for details on each category's rank. Overseas territories are included for each country when relevant. Countries in italics are common between the two ranks.

| country (no. of species) | threat rank | country (no. of species) | heritage rank |
|---|---|---|---|
| *Indonesia (32)* | 1 | Solomon Islands (9) | 1 |
| Argentina (26) | 2 | *Indonesia (32)* | 2 |
| *USA (20)* | 3 | Papua New Guinea (16) | 3 |
| *New Zealand (8)* | 4 | *United Kingdom (21)* | 4 |
| *Brazil (31)* | 5 | *New Zealand (8)* | 5 |
| *Madagascar (13)* | 6 | Australia (15) | 6 |
| *United Kingdom (21)* | 7 | *USA (20)* | 7 |
| Chile (13) | 8 | Philippines (16) | 8 |
| *Cuba (11)* | 9 | *Cuba (11)* | 9 |
| *Peru (27)* | 10 | *Japan (10)* | 10 |
| *Ecuador (23)* | 11 | *Madagascar (13)* | 11 |
| *Colombia (27)* | 12 | *Ecuador (23)* | 12 |
| *Venezuela (21)* | 13 | India (16) | 13 |
| *Japan (10)* | 14 | Seychelles (2) | 14 |
| Ethiopia (15) | 15 | *Venezuela (21)* | 15 |
| Bolivia (26) | 16 | *Brazil (31)* | 16 |
| *Mexico (17)* | 17 | *Colombia (27)* | 17 |
| Zimbabwe (17) | 18 | *Peru (27)* | 18 |
| South Africa (16) | 19 | *Mexico (17)* | 19 |
| Costa Rica (15) | 20 | France (30) | 20 |

having the most threatening consequences). Nevertheless, invasive species are not the main concern for other threatened island bird species globally, where over-exploitation and agriculture predominate [24]. The overwhelming threat posed by invasive predators to island rails could be linked to vulnerable adaptations like flightlessness, predator naivety [21,22,25] and their territorial lifestyle (e.g. ground-nesting), but more research is needed to disentangle rails' vulnerability in depth.

Here, we have demonstrated that one threatening pathway has not changed between extinct and threatened island rails: invasive predators are consistently the key problem (extinct sp.: 96%, threatened sp.: 62%), followed by over-hunting (extinct sp.: 54%, threatened sp.: 38%). Australasia/Oceania, where the problem of invasive predators is most prevalent, also has the most extinct and threatened rail species, and hosts 61% of the diversity of island endemic rails and 70% of all surviving flightless rails, making it the top one bioregion for conservation priority. However, Oceania, which in ancient times supported hundreds of flightless rails [9,10], now stands bleakly as a rail species graveyard after the mid-Holocene mass extinction (only five rail species survive in the Pacific basin, including just two endemics to the region).

On the other hand, 'Agriculture', 'Natural system modifications' (e.g. fire, dams and water abstraction) and 'Residential & commercial development' are the prevailing threats to continental rails, making the Neotropics the top two conservation priority bioregion. Compared with island rails, we found that continental rails were impacted by a wider diversity of threats. Indeed, 'Natural system modifications', 'Pollution' and 'Energy production & mining' are largely present for continental species, while mostly absent for island rails, leaving continental rails exposed, on average, to more threats than on islands. Moreover, threatening processes related to human population density [26] and land clearing [27,28] are found to impact continental bird species disproportionately in recent decades, compared with island species. Further research is required to identify the correlates to vulnerability in rails. To date, no rail

species have gone extinct on continents at human contact, but escalating contemporary threats could be creating another pathway in the threatening process to rails that could act to undermine their populations' resilience and so jeopardize the persistence of continental species in the long term, especially if the threats continue to intensify or act in synergy [24,29]. Only 11% of the continental rails are threatened, nevertheless, the IUCN Red List focusing on particular regions (such as the European Red List; https://www.iucnredlist.org/regions/europe) illustrates that some of the 'Least Concern' continental rails can appear as threatened at a smaller scale and, therefore, could be overlooked for conservation.

## 4.2. Conservation efforts, gaps and priorities

We found 'Research & Monitoring' effort to be the most important gap in rail conservation globally (25–42% of species in half the bioregions), with 'Ecosystem Protection & Management' almost as equally lacking (25–29% in half the bioregions). Likewise, de Lima *et al.* [30] found that the degree of research effort was biased across all bird species and areas. Moreover, threatened bird species tend to be less well-researched than non-threatened ones, probably due to their smaller geographical range [31]. Indeed, the Neotropics and Australasia/Oceania host the greatest number of threatened rail species and of threatened bird species in general, but both are understudied (0.03 and 0.08 published papers per threatened bird species, respectively, [31]).

Our analysis also revealed a positive outcome, as seven species improved their conservation status recently. An exemplar is the 30 years of conservation efforts, such as captive breeding and managed relocation, directed towards the Guam rail (*Hypotaenidia owstoni*), which improved from Extinct in the wild (EW) to CR. For the six other species (electronic supplementary material, table S5), the improvements in their conservation status all came from the discovery of more populations or a larger range, thanks to research surveys and monitoring. However, conservation actions included the protection of habitats in only a third of these species. This result highlights that more research and monitoring are needed to properly assess the conservation status of rails species, as it has been found for other taxa [32]. Hence, it appears as equally important to work on improving knowledge and protection for overlooked species, and especially for the two 'Data Deficient' rails, the brown-banded rail (*Lewinia mirifica*) and the Colombian crake (*Neocrex colombiana*). Moreover, the number of species whose IUCN status is deteriorating is increasing faster than the ones improving. In this context, an investigation into whether the temporal evolution of conservation status links to ongoing conservation actions for threatened species would be informative.

While Brooks *et al.* [31] found that most threatened bird species inhabit low-income countries, the top countries in our Threat rank for rails (the most threatened species and species with a declining IUCN status) include a mix of low- and high-income countries, with New Zealand and the USA in the top four. Low-income countries can be expected to carry threatened species as they typically have less enforcement and higher rates of wildlife poaching and habitat loss [4,8,33,34]. However, in the case of rails, high-income countries have more threatened species, because they possess, by chance, many islands supporting island endemic rails that are endangered due to different threatening mechanisms, such as invasive predators, that are not directly linked to a country's income. By analysing the threat patterns across spatial scales, we also uncovered important regional variation in the ranking of threats to rails (e.g. 'Natural system modifications' impacts 60% of the Palaearctic species, compared with only 17% globally), allowing us to address conservation priorities more specifically.

A few countries identified as rail 'conservation hotspots' (i.e. with high number of threatened rails and with high-conservation-value species) were also the ones most lacking in conservation efforts. In that regard, we suggest that Indonesia, the USA, the United Kingdom, New Zealand and Cuba should be focusing the most on measures to protect and recover their rail species, equally on their mainland and overseas territories. These countries are formed of islands or possess overseas territories (typically including many islands), on which rail species with heritage value (flightlessness or endemism) live. We suggest that these countries invest more conservation efforts in 'Research & Monitoring' and 'Ecosystem Protection & Management' (electronic supplementary material, table S4). The United Kingdom and New Zealand also need an increased 'Species Management' (electronic supplementary material, table S4).

The Solomon Islands is the country hosting the most flightless rails (four species, top two 'conservation hotspot', electronic supplementary material, table S7), and as only 19 flightless rails remain globally compared with the many hundreds existing in the Holocene [9,10], they should also be given particularly directed conservation attention. In particular, 'Research & Monitoring' are the most needed conservation actions for the Solomon species (electronic supplementary material, table S4).

Climate change also looms as a future threat for all island birds [35], especially via its impact on sea-level rise [10,36] and habitat loss through ecosystem degradation [12]. This would be particularly relevant

in the Pacific region that risks seeing the remaining island endemics wiped out under the impacts of climate change. As already found for the Hawaiian common gallinule (*Gallinula galeata sandvicensis* [37]), climate change's impacts can reduce habitat quality (and therefore an island's already limited carrying capacity), which can be especially threatening for rails endemic to small oceanic isolates—e.g. the Guam rail (*Hypotaenidia owstoni*), the Inaccessible Island rail (*Atlantisia rogersi*) and the Lord Howe woodhen (*Gallirallus sylvestris*).

Interestingly, while Foden *et al.* [38] found (based on their own framework) that 24–50% of all birds were globally vulnerable to climate change, we found only 14% of the rails species globally (26% of the island endemics) so impacted, based on the IUCN Red List framework. However, studies have suggested that the IUCN framework may be inadequate for detecting risks imposed by climate change on species [38–41].

Further research is needed to forecast the impacts of climate change on such restricted species in order to accurately determine the most adequate conservation measures to implement, such as defining climatic refugia, translocations or captivity plans [36,42–44].

## 5. Conclusion

The threat pattern in rails follows two different pathways. The world's extant island rails are continuing on the same trajectory as extinct species, with the majority of island endemics still suffering from the same key threats that drove others extinct (invasive predators and over-hunting). However, the synergy of modern threats has also created another trajectory, whereby continental rails are impacted by a diversity of contemporary anthropogenic activities that could jeopardize the long-term survival of previously resilient populations. This dichotomy leads to a complex pattern under which conservation options are branching into various profiles, whose priority will depend on the type of rail species (continental or island endemic) and the relevant scale (bioregions, countries and islands). This synthesis provides the first analysis of the spatio-temporal threat pattern in rails, and further research is needed to disentangle the role of extrinsic and intrinsic traits in threatening mechanisms, and thus better anticipate future rail extinctions.

Data accessibility. The data used in this article can be accessed at https://doi.org/10.5061/dryad.9s4mw6mfs [45].
Authors' contributions. All authors collaborated in conceiving and designing the study. L.L. collected the data, carried out the analyses and drafted the manuscript. J.C.B., S.C. and B.W.B. coordinated the study and critically revised the manuscript. All authors gave final approval for publication and agree to be held accountable for the work performed therein.
Competing interests. We declare we have no competing interests.
Funding. This research was funded by Australian Laureate Fellowship FL160100101 (Prof. Barry Brook).
Acknowledgements. We thank Stefania Ondei and Cristian Montalvo Mancheno for their help with ArcGIS and all individuals and organizations that participated in bird assessments for the IUCN Red List. We thank two anonymous reviewers for suggestions that greatly improved an earlier draft of the paper.

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
