## [Peer Review File · Royal Society Open Science]

Review History

RSOS-210262.R0 (Original submission)

Review form: Reviewer 1

Is the manuscript scientifically sound in its present form?

Yes

Are the interpretations and conclusions justified by the results?

Yes

Is the language acceptable?

Yes

Do you have any ethical concerns with this paper?

No

Have you any concerns about statistical analyses in this paper?

No

Recommendation?

Major revision is needed (please make suggestions in comments)

Comments to the Author(s)

I think that the manuscript has the potential to be published, since the authors evaluate important questions about threat patterns (spatially and temporally) of Rails and their conservation priorities and gaps. Although focused on a specific family of birds (Rallidae) such a study is valuable to inform and guide relevant conservation policy. Nevertheless, the manuscript needs some modifications before being published. I think that the most important parts to improve in the manuscript are Methods and Results. Some analysis (i.e. threat impact score) is not exploited and there is some inconsistency in the terms used (particularly threat names) that is somewhat confusing. See below for comments details.

L34: Maybe used "species' flight ability" instead of "species' unique evolutionary feature" which is less meaningful.

L57: Missing a point after "; Curnutt & Pimm, 2001").

L98-99: You made reference to islands endemic species L84-85, I advise you to move up this line in your paragraph.

L84-99: At the end of this paragraph, I advise you to indicate the total number of species and of threatened species used for the analyses.

L102-118: This paragraph contains a lot of redundant information which complicates your explanation. Here a suggest paragraph structure:

- Collection of threat information through IUCN; existence of different level of classification reflecting the complexity of the different anthropogenic pressures
- Computation of number of threats impacting threatened (and extinct, in view of your results section) species by using the second level of threat classification
- Computation of the Threat Impact Scoring System for all rail species by using the first level of threat classification.

L117-118: I do not understand why. Is it to not over-represented "biological resource use" threat?

L129-131: A notion not highlighted here is that is research and conservation actions needed for the taxa concerned. This is not obvious for people who do not know the IUCN classification well. They might think that you identify gaps in conservation efforts for rails based on conservation actions and research already setting-up. I advise you to give more details about the conservation actions classification scheme and research classification scheme.

L137-139: I am wondering how many times conservation status has been assessed for each species? If it is more than two times, how have you managed this?

L145: I would replace "with high conservation merit" by "with high conservation interest". For me, merit is not an appropriate term in a conservation objective.

L151: I would replace "unique evolutionary traits" by "a unique evolutionary trait", as you only considered the ability to flight if I understood correctly.

L153-154: I do not understand, when it would be relevant to classify species based on the different attributes. Please, explain this point.

Figure 1: Does a point refer to one threatened species? I count only 22 island endemic (not 21), does some points are identified for a same island or group of islands? In the same way, I count only 13 points of flightless instead of 14, does some points overlap and are hidden? In addition, I do not understand the localisation of some points, e.g. the one at the border of the USA and Mexico. Is it assigned to both countries or only to one, maybe the USA (with overseas territories) if points localisation is based on centroid. Please give more details in the legend of the figure and maybe change the localisation of points of countries composed by overseas territories in part.

L175: You did not consider these species as extinct in your study, and you removed them due to their uncertainty status. Please reformulate this part of sentence.

L176-177: Have you also some uncertainty about flight ability for some species alive? Maybe you can explain in methods section how you collect flight ability trait? What is the source(s) of the data?

L185: at global, continent or island scale?

L186-187: I do not understand what “(partly for 69%)” and “(partly for 62%)” refer to. This is not obvious at the reading that you make reference to island species. I understood it by looking your figure 2.

L187: What does “habitat loss” refer to? The threat terms are not consistent between the text and the Figure 2. Please be consistent in the terms used throughout your manuscript.

L190: “Threats due to agriculture were mostly found in the Afrotropics and the Neotropics” based on the number of species but if you look the percentage this are the Afrotropics and the Palearctic. Be clearer about this point maybe by reformulating the sentence (e.g., Rail species (in number) impacting by agriculture was mostly found in the Afrotropics and the Neotropics).

L185-198 and Figure 2: Why assess threat intensity and show it in figure 2 if you do not use this information in your results section. Add some lines about this point or remove it.

Review form: Reviewer 2

Is the manuscript scientifically sound in its present form?

Yes

Are the interpretations and conclusions justified by the results?

Yes

Is the language acceptable?

Yes

Do you have any ethical concerns with this paper?

No

Have you any concerns about statistical analyses in this paper?

No

Recommendation?

Major revision is needed (please make suggestions in comments)

Comments to the Author(s)

This is an interesting study that aims to perform a global synthesis of the temporal and spatial patterns of the threats to rail bird species (Aves: Rallidae) while identifying the current conservation priorities and gaps. Although the data used in the study is freely available on the IUCN Red List of Threatened Species, we believe that it is very useful to analyse, concentrate and discuss this information in a single document.

However, we have some issues regarding the methods, results, and discussion. Please find below comments on these issues, which we suggest that you take into consideration when reviewing the paper.

Major issues

1. We find that the methods are not always clear, especially due to the omission of important details of the data processing. We believe that, in its current version, the methods do not allow the readers to fully understand, repeat and validate the analyses.

At the beginning of the methods, the names of the sub-categories of threat appear to be different from those used by the IUCN Red List; sub-category 5.1 is named 'Hunting & collecting terrestrial animals' and not 'Hunting and direct exploitation', and 5.3 is named 'Logging & wood harvesting' instead of 'Logging & indirect effects' (lines 105 and 106).

In lines 115-117 you mention that you split category 5 in two sub-categories, 5.1 and 5.3. However, the script in the dryad repository ('Leveque_et_al._threats_global_and_island'), suggests that you used 5.1 and the sum of 5.2, 5.3, and 5.4, instead of 5.3. Is this correct? If so, we suggest that you clarify this in the text, since it will affect the interpretation of the results because the reader will not understand that those are related to three different sub-categories of threats (and not one).

Likewise, in the conservation status' analysis, you write "We gathered the possible classifications in five categories: 'Research, Monitoring and Planning', 'Ecosystem Protection and Management', 'Species Management', 'Education & Awareness', and 'Law & Policy' (i.e., legislative protection)" (lines 134 to 137), but you do not define those categories. By looking at the names of the categories in each scheme of the IUCN Red List, we can presume that: (i) 'Research, Monitoring and Planning' category was formed by grouping all categories in the 'Research Needed Classification Scheme'; (ii) 'Ecosystem Protection and Management' category was formed by grouping the first two categories, '1. Land/water protection' and '2. Land/water management', in the 'Conservation Actions Classification Scheme'; (iii) and the rest corresponded directly to each remaining category in the 'Conservation Actions Classification Scheme'. Furthermore, category 6 'Livelihood, economic & other incentives' of the 'Conservation Actions Classification Scheme' was not represented. The reason for this is not stated in the text, but one can assume that perhaps it did not appear in any of the studied species. These omissions impair the comprehension and reproducibility of the methods and, in our opinion, should be clarified.

This part of the methods would probably be clearer if you add a diagram illustrating how the various parameters were put together to obtain the results.

2. There is a lack of information regarding the methods used to collect the data for extinct rail species. In the section 'Spatial and temporal patterns of threats', the only mention to the threats of extinct rail species is at the end: "Finally, we compared the contemporary threats with the historical causes of extinctions to assess temporal changes in the threat pattern and distribution of threatened and extinct species." (lines 124 to 126). In the dryad repository, there is an excel named 'Extinct_rails_database_2020' with information about the threats of each species, but there are almost no references and the methods used for data collection are not indicated. In our opinion, this information should be included in the methods.

3. Figure 1 could be improved by keeping the circles from covering the islands completely. In some cases, it becomes impossible to identify the island, weakening the usefulness of the map. Inserting the circles near the islands and connecting them with a line or arrow would avoid this problem, making the figure more informative.

4. You state that "Threats due to agriculture were mostly found in the Afrotropics and the Neotropics" (line 190). But Figure 2B indicates that agriculture was also considered a threat in 40% of the Palearctic rail species. It is not clear why you highlight the Neotropics, where only ~21-22% of the species were impacted by agriculture?

5. The discussion and conclusions of the manuscript lack the conservation focus that is promised in the goals. Currently, the discussion is largely a description of the results and is missing an actual debate on their implications to the conservation of rail bird species worldwide, or to the planning of specific conservation actions. For example, a discussion about why seven

species recently improved their conservation status could provide clues on what kind of improvements should be made.

Furthermore, in the abstract you indicate that “Forecasting the impacts of climate change on island endemic rails and disentangling the specific roles of extrinsic and intrinsic traits (like flightlessness), will be particularly valuable avenues of research for improving our forecasts of rail vulnerability.”. However, in the discussion, you do not elaborate on why climate change did not appear to be an important threat for rail species. A short discussion of this result would enrich this paragraph (lines 343 to 356).

Likewise, concerning the countries identified both as rail ‘conservation hotspots’ and those most lacking in conservation efforts (i.e., Indonesia, the U.S.A., the United Kingdom, New Zealand and Cuba), you do not propose a justification for this result, or define specific actions to conserve rail species: “In that regard, we suggest that Indonesia, the U.S.A., the United Kingdom, New Zealand, and Cuba should be focusing more on measures to protect and recover their rail species, equally on their mainland and overseas territories.” (lines 336 to 338).

Finally, the discussion can be enriched by adding more references to support some of the explanations used to corroborate the results. For example, to justify why continental rails appear to be impacted by a wider diversity of threats, you merely say that this could be due to these species “being intrinsically more resistant to threats, even in synergy, or because fewer threats might naturally occur on islands” (lines 293 and 294). However, why would islands have fewer threats? Maybe you can explain this adding references and examples of threats that are less intense on islands.

Minor issues

1. Please write the word “family” in lowercase (line 21).
2. For the sake of consistency, we suggest that you always use either e.g. or e.g.,
3. Please add a final point after the parenthesis (line 57).
4. Improve coherence by changing the comma after “programs,” to a semicolon (line 77), since it is the symbol used to separate the first and second goals.
5. To improve clarity, you may want to add ‘so’ to the following sentence: “Similarly, the New Caledonian rail (*Gallirallus lafresnayanus*) and the Samoan moorhen (*Pareudiastes pacificus*) are two ‘Critically endangered’ rail species that have not been seen with certainty since the 19th century and are suspected to be extinct (IUCN, 2019), so they were also excluded from our analysis (Table S2)” (lines 94 to 98).
6. Throughout the manuscript, five different spatial scales/levels are used across analyses: global (considering all continents and islands), continents (considering only continents), islands (considering only islands), bioregions (considering bioregions defined by Olson et al., 2001), and countries (used in the analyses focused on ‘Conservation status and gaps’). The global scale is defined as ‘worldwide’ and ‘globally’, and the bioregions as ‘bioregions’ and ‘bioregionally’, which may make things less clear. We suggest that you use the same term to designate each level in the entire manuscript.
7. Please add the number of threats considered in the analysis of threat diversity (lines 102 to 107). If you used the first sub-category of threat as the level of threat and there are 45 sub-categories according to the IUCN Red List of Threatened Species, this means that the maximum number of threats for one species is 45, correct?
8. You should remove the comma after “severity,” and add one after “timing” (line 110).

9. Indicate which version of the 'Threats Classification Scheme' was used in section 'Spatial and temporal patterns of threats'.
10. Please clarify that only two of the three levels considered for the timing of the threat were used in the analyses (past and ongoing). This information can be added, for example, in the following sentence "We included 'Past' impact to illustrate the temporal evolution of threats" (lines 112 and 113).
11. To use the same names throughout the manuscript (following the methods described in lines 103 and 104), we suggest that you rewrite the text "At this level, we used the first level of threat" to "For this analysis, we used the first category as the level of threat" (line 114).
12. To increase comprehension, we suggest that you change the text to "whether or not they were considered threatened by IUCN" (lines 119 and 120).
13. The "Research classification scheme" (line 130) corresponds to the 'Research Needed Classification Scheme'? If yes, please replace it with its correct name and also include the accessed version.
14. Remove "being" in "five were not being documented sufficiently to be confident about their flight ability" (lines 176 and 177).
15. We suggest that you include the total number of species (n = 140) considered in your database of threatened rail species, since it is the first time this database is mentioned in the results (line 178).
16. In the legend of Table 1 please add an "s" to "scale" (...scales). Moreover, the table represents the number of threats and not the proportion of threats, right? We also suggest that you include the range of values considering all threatened rail species, that is the minimum and the maximum number of threats found.
17. In Table 1, the median of the continental threatened rails is 4, and the first quartile is also 4. Please check if this is correct.
18. Looking at your files 'threat_type_database2019' and 'threats_region2019', we assume that the sub-category 10 named 'Geological events' was not listed for any species because it was not listed in any of them. If this is the case, please add a sentence clarifying this issue in the results' section 'Spatial and temporal pattern of threats'.
19. To improve the clarity of Figure 2 and maintain the coherence among all figures we suggest that you: rewrite the description of panel (A) to match panel (B): "Threat diversity and intensity, measured as the impact score proposed by the IUCN Red List of Threatened Species, for rail species at a global scale (Gb; n = 140), for island endemic rail species (Is; n = 42), and for continental rail species (Ct; n = 98)." (lines 207 and 208); change "per category" to "per bioregion" (line 210). Lastly, we suggest that the y-axis is renamed to "percentage of rail species impacted by threat".
20. Still concerning Figure 2, the first panel (A) also does not include the categories "Other" and "Energy production and mining". Please explain this as you did for panel (B).
21. Remove "proportionally" in the following sentence: "'Pollution' and 'Natural system modifications' (e.g., fire, dams, water abstraction) were the only threats that were proportionally more threatening on continents than for island endemics (Fig. 2A)." to "'Pollution' and 'Natural

system modifications' (e.g., fire, dams, water abstraction) were the only threats that were more threatening for continental rails than for island endemics (Fig. 2A)." (lines 194 to 196).

22. We suggest that you change the location of the word "globally" and specify the panel of Figure 2: "Globally, less than 20% of rails are impacted by 'Natural System modifications', however, it is an important factor in the Palearctic, where it impacts 60% of that region's rails (Fig. 2B)." (lines 196 to 198).

23. It is unclear why the category "Research, Monitoring and Planning" was named "Research & monitoring" in line 212. Are you considering the categories 'Research' and 'Monitoring'? If yes, why would you say "We gathered the possible classifications in five categories" (lines 134 and 135)? Please clarify this issue to avoid confusion.

24. To improve clarity, we suggest that you rewrite the following sentence "including six species that became newly threatened" to "including six species that were again classified as threatened" (lines 226 and 227), and add "Out of those 11 species," before "Eight species were endemic...", and remove "species" after "Eight..." (line 227).

25. Add a reference to this statement: "Most of the past rail extinctions occurred on the Pacific islands, directly linked to the fact that they were the support for the largest radiation of rails, reaching high levels of endemism." (lines 258 to 260).

26. We suggest that you remove "in general" (line 261) and reword the sentence to: "Flightlessness is an evolutionary trait that has been found to make bird species more extinction-prone during different waves of extinction (Duncan et al., 2002; Boyer, 2008), especially rails (Steadman, 1995; Curnutt & Pimm, 2001), and it also appears as an important contributor to the vulnerability of rails to contemporary threatening processes."

27. You may want to replace "globally" with 'worldwide' to avoid using the same word many times (line 270).

28. Provide references for the following affirmation "However, Oceania, which in ancient times supported hundreds of flightless rails, now stands bleakly as a rail-species graveyard after the Holocene mass extinction (only five rail species survive in the Pacific basin, including just two endemics to the region)." (lines 285 to 288).

29. We find the font used in Figure 2 a bit too small to be easily readable.

Decision letter (RSOS-210262.R0)

Dear Miss Lévêque,

The Editors assigned to your paper RSOS-210262 "Characterising the spatio-temporal threats, conservation hotspots, and conservation gaps for the most extinction-prone bird family (Aves: Rallidae)" have now received comments from reviewers and would like you to revise the paper in accordance with the reviewer comments and any comments from the Editors. Please note this decision does not guarantee eventual acceptance.

Please submit your revised manuscript and required files (see below) no later than 21 days from today's (ie 28-Jun-2021) date. Note: the ScholarOne system will 'lock' if submission of the revision is attempted 21 or more days after the deadline. If you do not think you will be able to meet this deadline please contact the editorial office immediately.

on behalf of Dr Joachim Mergeay (Associate Editor) and Pete Smith (Subject Editor)
openscience@royalsociety.org

Associate Editor Comments to Author (Dr Joachim Mergeay):

Dear Dr. Lévêque,

First of all, please accept our apologies for the delay we have had with your manuscript. We've had a hard time finding suitable and willing reviewers.

We have now received two constructive yet critical reviews on your manuscript. Both reviewers agree that the topic is broadly well addressed, and that it makes good use of publicly available data. However, they also agree that certain aspects are not clear as to how certain analyses were conducted and provide several important suggestions for improvement of the manuscript, across all sections.

Sincerely,
Joachim Mergeay

Reviewer comments to Author:

Reviewer: 1

Comments to the Author(s)

I think that the manuscript has the potential to be published, since the authors evaluate important questions about threat patterns (spatially and temporally) of Rails and their conservation

priorities and gaps. Although focused on a specific family of birds (Rallidae) such a study is valuable to inform and guide relevant conservation policy. Nevertheless, the manuscript needs some modifications before being published. I think that the most important parts to improve in the manuscript are Methods and Results. Some analysis (i.e. threat impact score) is not exploited and there is some inconsistency in the terms used (particularly threat names) that is somewhat confusing. See below for comments details.

L34: Maybe used “species’ flight ability” instead of “species’ unique evolutionary feature” which is less meaningful.

L57: Missing a point after “; Curnutt & Pimm, 2001)”.

L98-99: You made reference to islands endemic species L84-85, I advise you to move up this line in your paragraph.

L84-99: At the end of this paragraph, I advise you to indicate the total number of species and of threatened species used for the analyses.

L102-118: This paragraph contains a lot of redundant information which complicates your explanation. Here a suggest paragraph structure:

- Collection of threat information through IUCN; existence of different level of classification reflecting the complexity of the different anthropogenic pressures
- Computation of number of threats impacting threatened (and extinct, in view of your results section) species by using the second level of threat classification
- Computation of the Threat Impact Scoring System for all rail species by using the first level of threat classification.

L117-118: I do not understand why. Is it to not over-represented “biological resource use” threat?

L129-131: A notion not highlighted here is that is research and conservation actions needed for the taxa concerned. This is not obvious for people who do not know the IUCN classification well. They might think that you identify gaps in conservation efforts for rails based on conservation actions and research already setting-up. I advise you to give more details about the conservation actions classification scheme and research classification scheme.

L137-139: I am wondering how many times conservation status has been assessed for each species? If it is more than two times, how have you managed this?

L145: I would replace “with high conservation merit” by “with high conservation interest”. For me, merit is not an appropriate term in a conservation objective.

L151: I would replace “unique evolutionary traits” by “a unique evolutionary trait”, as you only considered the ability to flight if I understood correctly.

L153-154: I do not understand, when it would be relevant to classify species based on the different attributes. Please, explain this point.

Figure 1: Does a point refer to one threatened species? I count only 22 island endemic (not 21), does some points are identified for a same island or group of islands? In the same way, I count only 13 points of flightless instead of 14, does some points overlap and are hidden? In addition, I do not understand the localisation of some points, e.g. the one at the border of the USA and Mexico. Is it assigned to both countries or only to one, maybe the USA (with overseas territories) if points localisation is based on centroid. Please give more details in the legend of the figure and maybe change the localisation of points of countries composed by overseas territories in part.

L175: You did not consider these species as extinct in your study, and you removed them due to their uncertainty status. Please reformulate this part of sentence.

L176-177: Have you also some uncertainty about flight ability for some species alive? Maybe you can explain in methods section how you collect flight ability trait? What is the source(s) of the data?

L185: at global, continent or island scale?

L186-187: I do not understand what “(partly for 69%)” and “(partly for 62%)” refer to. This is not obvious at the reading that you make reference to island species. I understood it by looking your figure 2.

L187: What does “habitat loss” refer to? The threat terms are not consistent between the text and the Figure 2. Please be consistent in the terms used throughout your manuscript.

L190: “Threats due to agriculture were mostly found in the Afrotropics and the Neotropics” based on the number of species but if you look the percentage this are the Afrotropics and the Palearctic. Be clearer about this point maybe by reformulating the sentence (e.g., Rail species (in number) impacting by agriculture was mostly found in the Afrotropics and the Neotropics).

L185-198 and Figure 2: Why assess threat intensity and show it in figure 2 if you do not use this information in your results section. Add some lines about this point or remove it.

Reviewer: 2

Comments to the Author(s)

This is an interesting study that aims to perform a global synthesis of the temporal and spatial patterns of the threats to rail bird species (Aves: Rallidae) while identifying the current conservation priorities and gaps. Although the data used in the study is freely available on the IUCN Red List of Threatened Species, we believe that it is very useful to analyse, concentrate and discuss this information in a single document.

However, we have some issues regarding the methods, results, and discussion. Please find below comments on these issues, which we suggest that you take into consideration when reviewing the paper.

Major issues

1. We find that the methods are not always clear, especially due to the omission of important details of the data processing. We believe that, in its current version, the methods do not allow the readers to fully understand, repeat and validate the analyses.

At the beginning of the methods, the names of the sub-categories of threat appear to be different from those used by the IUCN Red List; sub-category 5.1 is named ‘Hunting & collecting terrestrial animals’ and not ‘Hunting and direct exploitation’, and 5.3 is named ‘Logging & wood harvesting’ instead of ‘Logging & indirect effects’ (lines 105 and 106).

In lines 115-117 you mention that you split category 5 in two sub-categories, 5.1 and 5.3. However, the script in the dryad repository (`‘Leveque_et_al._threats_global_and_island’`), suggests that you used 5.1 and the sum of 5.2, 5.3, and 5.4, instead of 5.3. Is this correct? If so, we suggest that you clarify this in the text, since it will affect the interpretation of the results because the reader will not understand that those are related to three different sub-categories of threats (and not one).

Likewise, in the conservation status’ analysis, you write “We gathered the possible classifications in five categories: ‘Research, Monitoring and Planning’, ‘Ecosystem Protection and Management’, ‘Species Management’, ‘Education & Awareness’, and ‘Law & Policy’ (i.e., legislative protection)” (lines 134 to 137), but you do not define those categories. By looking at the names of the categories in each scheme of the IUCN Red List, we can presume that: (i) ‘Research, Monitoring and Planning’ category was formed by grouping all categories in the ‘Research Needed Classification Scheme’; (ii) ‘Ecosystem Protection and Management’ category was formed by grouping the first two categories, ‘1. Land/water protection’ and ‘2. Land/water management’, in the ‘Conservation Actions Classification Scheme’; (iii) and the rest corresponded directly to each remaining category in the ‘Conservation Actions Classification Scheme’. Furthermore, category 6 ‘Livelihood, economic & other incentives’ of the ‘Conservation Actions Classification Scheme’ was not represented. The reason for this is not stated in the text, but one can assume that perhaps it did not appear in any of the studied species. These omissions impair the comprehension and reproducibility of the methods and, in our opinion, should be clarified.

This part of the methods would probably be clearer if you add a diagram illustrating how the various parameters were put together to obtain the results.

2. There is a lack of information regarding the methods used to collect the data for extinct rail species. In the section 'Spatial and temporal patterns of threats', the only mention to the threats of extinct rail species is at the end: "Finally, we compared the contemporary threats with the historical causes of extinctions to assess temporal changes in the threat pattern and distribution of threatened and extinct species." (lines 124 to 126). In the dryad repository, there is an excel named 'Extinct_rails_database_2020' with information about the threats of each species, but there are almost no references and the methods used for data collection are not indicated. In our opinion, this information should be included in the methods.

3. Figure 1 could be improved by keeping the circles from covering the islands completely. In some cases, it becomes impossible to identify the island, weakening the usefulness of the map. Inserting the circles near the islands and connecting them with a line or arrow would avoid this problem, making the figure more informative.

4. You state that "Threats due to agriculture were mostly found in the Afrotropics and the Neotropics" (line 190). But Figure 2B indicates that agriculture was also considered a threat in 40% of the Palearctic rail species. It is not clear why you highlight the Neotropics, where only ~21-22% of the species were impacted by agriculture?

5. The discussion and conclusions of the manuscript lack the conservation focus that is promised in the goals. Currently, the discussion is largely a description of the results and is missing an actual debate on their implications to the conservation of rail bird species worldwide, or to the planning of specific conservation actions. For example, a discussion about why seven species recently improved their conservation status could provide clues on what kind of improvements should be made.

Furthermore, in the abstract you indicate that "Forecasting the impacts of climate change on island endemic rails and disentangling the specific roles of extrinsic and intrinsic traits (like flightlessness), will be particularly valuable avenues of research for improving our forecasts of rail vulnerability.". However, in the discussion, you do not elaborate on why climate change did not appear to be an important threat for rail species. A short discussion of this result would enrich this paragraph (lines 343 to 356).

Likewise, concerning the countries identified both as rail 'conservation hotspots' and those most lacking in conservation efforts (i.e., Indonesia, the U.S.A., the United Kingdom, New Zealand and Cuba), you do not propose a justification for this result, or define specific actions to conserve rail species: "In that regard, we suggest that Indonesia, the U.S.A., the United Kingdom, New Zealand, and Cuba should be focusing more on measures to protect and recover their rail species, equally on their mainland and overseas territories." (lines 336 to 338).

Finally, the discussion can be enriched by adding more references to support some of the explanations used to corroborate the results. For example, to justify why continental rails appear to be impacted by a wider diversity of threats, you merely say that this could be due to these species "being intrinsically more resistant to threats, even in synergy, or because fewer threats might naturally occur on islands" (lines 293 and 294). However, why would islands have fewer threats? Maybe you can explain this adding references and examples of threats that are less intense on islands.

Minor issues

1. Please write the word "family" in lowercase (line 21).
2. For the sake of consistency, we suggest that you always use either e.g. or e.g.,
3. Please add a final point after the parenthesis (line 57).

4. Improve coherence by changing the comma after “programs,” to a semicolon (line 77), since it is the symbol used to separate the first and second goals.
5. To improve clarity, you may want to add ‘so’ to the following sentence: “Similarly, the New Caledonian rail (*Gallirallus lafresnayanus*) and the Samoan moorhen (*Pareudiastes pacificus*) are two ‘Critically endangered’ rail species that have not been seen with certainty since the 19th century and are suspected to be extinct (IUCN, 2019), so they were also excluded from our analysis (Table S2)” (lines 94 to 98).
6. Throughout the manuscript, five different spatial scales/levels are used across analyses: global (considering all continents and islands), continents (considering only continents), islands (considering only islands), bioregions (considering bioregions defined by Olson et al., 2001), and countries (used in the analyses focused on ‘Conservation status and gaps’). The global scale is defined as ‘worldwide’ and ‘globally’, and the bioregions as ‘bioregions’ and ‘bioregionally’, which may make things less clear. We suggest that you use the same term to designate each level in the entire manuscript.
7. Please add the number of threats considered in the analysis of threat diversity (lines 102 to 107). If you used the first sub-category of threat as the level of threat and there are 45 sub-categories according to the IUCN Red List of Threatened Species, this means that the maximum number of threats for one species is 45, correct?
8. You should remove the comma after “severity,” and add one after “timing” (line 110).
9. Indicate which version of the ‘Threats Classification Scheme’ was used in section ‘Spatial and temporal patterns of threats’.
10. Please clarify that only two of the three levels considered for the timing of the threat were used in the analyses (past and ongoing). This information can be added, for example, in the following sentence “We included ‘Past’ impact to illustrate the temporal evolution of threats” (lines 112 and 113).
11. To use the same names throughout the manuscript (following the methods described in lines 103 and 104), we suggest that you rewrite the text “At this level, we used the first level of threat” to “For this analysis, we used the first category as the level of threat” (line 114).
12. To increase comprehension, we suggest that you change the text to “whether or not they were considered threatened by IUCN” (lines 119 and 120).
13. The “Research classification scheme” (line 130) corresponds to the ‘Research Needed Classification Scheme’? If yes, please replace it with its correct name and also include the accessed version.
14. Remove “being” in “five were not being documented sufficiently to be confident about their flight ability” (lines 176 and 177).
15. We suggest that you include the total number of species ($n = 140$) considered in your database of threatened rail species, since it is the first time this database is mentioned in the results (line 178).
16. In the legend of Table 1 please add an “s” to “scale” (...scales). Moreover, the table represents the number of threats and not the proportion of threats, right? We also suggest that you include

the range of values considering all threatened rail species, that is the minimum and the maximum number of threats found.

17. In Table 1, the median of the continental threatened rails is 4, and the first quartile is also 4. Please check if this is correct.

18. Looking at your files 'threat_type_database2019' and 'threats_region2019', we assume that the sub-category 10 named 'Geological events' was not listed for any species because it was not listed in any of them. If this is the case, please add a sentence clarifying this issue in the results' section 'Spatial and temporal pattern of threats'.

19. To improve the clarity of Figure 2 and maintain the coherence among all figures we suggest that you: rewrite the description of panel (A) to match panel (B): "Threat diversity and intensity, measured as the impact score proposed by the IUCN Red List of Threatened Species, for rail species at a global scale (Gb; n = 140), for island endemic rail species (Is; n = 42), and for continental rail species (Ct; n = 98)." (lines 207 and 208); change "per category" to "per bioregion" (line 210). Lastly, we suggest that the y-axis is renamed to "percentage of rail species impacted by threat".

20. Still concerning Figure 2, the first panel (A) also does not include the categories "Other" and "Energy production and mining". Please explain this as you did for panel (B).

21. Remove "proportionally" in the following sentence: "'Pollution' and 'Natural system modifications' (e.g., fire, dams, water abstraction) were the only threats that were proportionally more threatening on continents than for island endemics (Fig. 2A)." to "'Pollution' and 'Natural system modifications' (e.g., fire, dams, water abstraction) were the only threats that were more threatening for continental rails than for island endemics (Fig. 2A)." (lines 194 to 196).

22. We suggest that you change the location of the word "globally" and specify the panel of Figure 2: "Globally, less than 20% of rails are impacted by 'Natural System modifications', however, it is an important factor in the Palearctic, where it impacts 60% of that region's rails (Fig. 2B)." (lines 196 to 198).

23. It is unclear why the category "Research, Monitoring and Planning" was named "Research & monitoring" in line 212. Are you considering the categories 'Research' and 'Monitoring'? If yes, why would you say "We gathered the possible classifications in five categories" (lines 134 and 135)? Please clarify this issue to avoid confusion.

24. To improve clarity, we suggest that you rewrite the following sentence "including six species that became newly threatened" to "including six species that were again classified as threatened" (lines 226 and 227), and add "Out of those 11 species," before "Eight species were endemic...", and remove "species" after "Eight..." (line 227).

25. Add a reference to this statement: "Most of the past rail extinctions occurred on the Pacific islands, directly linked to the fact that they were the support for the largest radiation of rails, reaching high levels of endemism." (lines 258 to 260).

26. We suggest that you remove "in general" (line 261) and reword the sentence to: "Flightlessness is an evolutionary trait that has been found to make bird species more extinction-prone during different waves of extinction (Duncan et al., 2002; Boyer, 2008), especially rails (Steadman, 1995; Curnutt & Pimm, 2001), and it also appears as an important contributor to the vulnerability of rails to contemporary threatening processes."

27. You may want to replace “globally” with ‘worldwide’ to avoid using the same word many times (line 270).

28. Provide references for the following affirmation “However, Oceania, which in ancient times supported hundreds of flightless rails, now stands bleakly as a rail-species graveyard after the Holocene mass extinction (only five rail species survive in the Pacific basin, including just two endemics to the region).” (lines 285 to 288).

29. We find the font used in Figure 2 a bit too small to be easily readable.

===PREPARING YOUR MANUSCRIPT===

===PREPARING YOUR REVISION IN SCHOLARONE===

Please ensure that you include a summary of your paper at Step 2 'Type, Title, & Abstract'. This should be no more than 100 words to explain to a non-scientific audience the key findings of your

research. This will be included in a weekly highlights email circulated by the Royal Society press office to national UK, international, and scientific news outlets to promote your work.

Author's Response to Decision Letter for (RSOS-210262.R0)

See Appendix A.

Decision letter (RSOS-210262.R1)

Dear Miss Lévêque,

It is a pleasure to accept your manuscript entitled "Characterising the spatio-temporal threats, conservation hotspots, and conservation gaps for the most extinction-prone bird family (Aves: Rallidae)" in its current form for publication in Royal Society Open Science. The comments of the Editors are included at the foot of this letter.

on behalf of Dr Joachim Mergeay (Associate Editor) and Pete Smith (Subject Editor)
openscience@royalsociety.org

Associate Editor Comments to Author (Dr Joachim Mergeay):
Dear Miss Lévêque,

Thank you for your resubmission.

The comments of the reviewers were addressed in a consistent manner, and this has clearly improved the manuscript.

The office and production team will fill you in on the details.

Best regards,
Joachim Mergeay

Appendix A

RESPONSES TO COMMENTS

Dear Dr Joachim Mergeay,

On behalf of all co-authors, I thank you for giving us the opportunity to submit a revised version of our manuscript RSOS-210262, titled "**Characterising the spatio-temporal threats, conservation hotspots, and conservation gaps for the most extinction-prone bird family (Aves: Rallidae)**" to *Royal Society Open Science*. We are grateful for your efforts to find referees and appreciate the quality of the referees' comments. We endeavored to revise our manuscript while addressing the excellent points made by both referees and incorporating their suggestions. We thank you for your vote of confidence.

Below is a point-by-point response to the comments, and those raised by the referees.

Please note that the quoted lines refer to the 'tracked-changes' version of our manuscript.

COMMENTS FROM REFEREE 1	
Comment 1:	I think that the manuscript has the potential to be published, since the authors evaluate important questions about threat patterns (spatially and temporally) of Rails and their conservation priorities and gaps. Although focused on a specific family of birds (Rallidae) such a study is valuable to inform and guide relevant conservation policy. Nevertheless, the manuscript needs some modifications before being published. I think that the most important parts to improve in the manuscript are Methods and Results. Some analysis (i.e. threat impact score) is not exploited and there is some inconsistency in the terms used (particularly threat names) that is somewhat confusing. See below for comments details.
Response:	We thank you for your favorable assessment about the relevance of our study to the journal. We took on board your overall concerns and specific comments, focusing particularly on clarifying Methods and Results (see response to comments below).
Comment 2:	L34: Maybe used "species' flight ability" instead of "species' unique evolutionary feature" which is less meaningful.
Response:	We agree with the referee that the term was not adequate. Nevertheless, this part of the abstract has been removed to meet the word limit (200 words).
Comment 3:	L57: Missing a point after "; Curnutt & Pimm, 2001)".
Response:	We accepted this suggestion (line 58).
Comment 4:	L98-99: You made reference to islands endemic species L84-85, I advise you to move up this line in your paragraph.
Response:	We moved this section as suggested (line 95).
Comment 5:	L84-99: At the end of this paragraph, I advise you to indicate the total number of species and of threatened species used for the analyses.
Response:	We accepted this suggestion, adding: "We compiled a database for all 144 species of extant rails (including 42 island endemic species and 33 threatened species)" (line 86).
Comment 6:	L102-118: This paragraph contains a lot of redundant information which complicates your explanation. Here a suggest paragraph structure: - Collection of threat information through IUCN; existence of different level of classification reflecting the complexity of the different anthropogenic pressures - Computation of number of threats impacting threatened (and extinct, in view

Response:	of your results section) species by using the second level of threat classification - Computation of the Threat Impact Scoring System for all rail species by using the first level of threat classification. We agree that this paragraph could be restructured to improve understanding. We have revised the manuscript by restructuring the key analyses and by adding sub-headings to the different parts of the methods (lines 105-150) and related results. This has led to a more efficient and logical presentation.
Comment 7: Response:	L117-118: I do not understand why. Is it to not over-represented "biological resource use" threat? While the two types of threats ('Hunting & collecting terrestrial animals' and 'Logging & indirect effects') are considered by the IUCN under the broad category 'Biological resource use', we believe that they are quite distinct in the mechanisms – one being a direct action of the birds' removal through hunting, while the other is an indirect consequence, related mostly to deforestation. These two types of threats are, remarkably, among the most important threats to rails. Their mechanisms and origins (type of human actions) are quite distinct. As our paper has a goal of influencing conservation policy, we believe that it is important and more informative to visually separate these two types of threatening processes. This breakdown has also been practiced previously in the literature, e.g.,: Olah, G., Butchart, S.H.M., Symes, A., Guzmán, I.M., Cunningham, R., Brightsmith, D.J. & Heinsohn, R. (2016) Ecological and socio-economic factors affecting extinction risk in parrots. Biodiversity and Conservation, 25, 205-223. https://doi.org/10.1007/s10531-015-1036-z For improved clarity, in the revision we incorporated further details on what sub-categories were included in the separation: "We split the threat '5. Biological resource use' in two categories: 'Hunting and collecting terrestrial animals' (5.1.) and 'Logging & indirect effects' (regrouping '5.2. Gathering terrestrial plants', '5.3 Logging and wood harvesting', and '5.4. Fishing & harvesting aquatic resources')". (lines 145-149).
Comment 8: Response:	L129-131: A notion not highlighted here is that is research and conservation actions needed for the taxa concerned. This is not obvious for people who do not know the IUCN classification well. They might think that you identify gaps in conservation efforts for rails based on conservation actions and research already setting-up. I advise you to give more details about the conservation actions classification scheme and research classification scheme. Lines 170-172, we explained how research and actions needed come from the IUCN schemes. To clarify further, we added that research and actions needed are also identified by the IUCN (line 171).
Comment 9: Response:	L137-139: I am wondering how many times conservation status has been assessed for each species? If it is more than two times, how have you managed this? We have now clarified this part of our methods in lines 181-187: "Changes in conservation status could happen any year between 1988 and 2019. In cases where status changed more than once, we only used the most recent change."
Comment 10:	L145: I would replace "with high conservation merit" by "with high conservation interest". For me, merit is not an appropriate term in a conservation objective.

Response:	We accepted this suggestion (line 201).
Comment 11:	L151: I would replace “unique evolutionary traits” by “a unique evolutionary trait”, as you only considered the ability to flight if I understood correctly.
Response:	We accepted this suggestion (line 207).
Comment 12:	L153-154: I do not understand, when it would be relevant to classify species based on the different attributes. Please, explain this point.
Response:	We developed our explanations by adding two examples to improve understanding of our methods (lines 210-212).
Comment 13:	Figure 1: Does a point refer to one threatened species? I count only 22 island endemic (not 21), does some points are identified for a same island or group of islands? In the same way, I count only 13 points of flightless instead of 14, does some points overlap and are hidden? In addition, I do not understand the localisation of some points, e.g. the one at the border of the USA and Mexico. Is it assigned to both countries or only to one, maybe the USA (with overseas territories) if points localisation is based on centroid. Please give more details in the legend of the figure and maybe change the localisation of points of countries composed by overseas territories in part.
Response:	As the referee noticed, there were some inconsistencies in the map. Some threatened species had an update of their IUCN assessment after the map was initially created, and while we updated all our analyses, we neglected to update the map. In the revised version, we have adjusted the map to incorporate these changes and updated their location to be at the centroid of the species' range, as found on their IUCN profiles. We edited the legend to include this information.
Comment 14:	L175: You did not consider these species as extinct in your study, and you removed them due to their uncertainty status. Please reformulate this part of sentence.
Response:	We agree with the referee that this part of our results required more clarification. The two species were removed from the study on extant species because they are considered as extinct and were therefore included in the analyses with extinct species. We made a modification in our methods (line 100) and supplementary material (Table S2) to improve clarity.
Comment 15:	L176-177: Have you also some uncertainty about flight ability for some species alive? Maybe you can explain in methods section how you collect flight ability trait? What is the source(s) of the data?
Response:	There were uncertainties about flight ability for some extinct species as their adaptations are inferred only from fossils or sub-fossils. Flightlessness is a multifactorial evolutionary process and can involve various morphological changes. In some instances, this made it hard to extrapolate with confidence regarding their ability to fly. As mentioned in line 85, all information on rail species were taken from Taylor & van Perlo (1998) and IUCN (2019). To underscore the evidence base for this classification, we cited these references again in the results (lines 239-240).
Comment 16:	L185: at global, continent or island scale?
Response:	As we previously stated line 236 “All recent rail extinctions (post 1500 A.D.) occurred on islands”, such that in line 256: “Introduced predators and over-hunting were the main drivers to rail extinctions:”, we assumed that there was no need to restate this. However, as this is an important finding, we have now reiterated the origin of the extinctions (line 256) as suggested by the referee.

Comment 17:	186-187: I do not understand what “(partly for 69%)” and “(partly for 62%)” refer to. This is not obvious at the reading that you make reference to island species. I understood it by looking your figure 2.
Response:	We agree with the referee’s critique that this part of our results could be confusing or misinterpreted. As such, we rewrote this sentence to better explain that these threats can be involved with other threats for X% of the extinctions (lines 257-260). As stated in our response to comment 16, we have also restated that the extinctions were from islands (line 256). However, Figure 2 does not include extinct species.
Comment 18:	L187: What does “habitat loss” refer to? The threat terms are not consistent between the text and the Figure 2. Please be consistent in the terms used throughout your manuscript.
Response:	As we initially simplified the categorization of causes of extinctions into three main categories, however we agree that the use of consistent terminology would improve the manuscript. In lines 260-264, we defined “habitat loss” as the official ICUN categories: “‘Natural system modifications’ (‘fire & fire suppression’, ‘other ecosystem modifications’), ‘Agriculture & aquaculture’ (‘livestock farming & ranching’, ‘annual & perennial non-timber crops’), ‘Invasive and other problematic species, genes & diseases’, and ‘Climate change & severe weather’ (‘storms & flooding’)”
Comment 19:	L190: “Threats due to agriculture were mostly found in the Afrotropics and the Neotropics” based on the number of species but if you look the percentage this are the Afrotropics and the Palearctic. Be clearer about this point maybe by reformulating the sentence (e.g., Rail species (in number) impacting by agriculture was mostly found in the Afrotropics and the Neotropics).
Response:	We thank the referee for raising this issue, and have accepted this suggestion (line 275).
Comment 20:	L185-198 and Figure 2: Why assess threat intensity and show it in figure 2 if you do not use this information in your results section. Add some lines about this point or remove it.
Response:	We agree and have revised the results to improve this section. In lines 268-270, we added: “No threats were found to be of ‘High impact’ (‘Rapid’ or ‘Very rapid’ severity, combined with ‘Majority’ or ‘Whole’ Scope). ‘Invasive & problematic species’, and ‘Climate change & severe weather’ were the most prevalent threats with a medium impact.”

COMMENTS FROM REFEREE 2

Comment 1:	This is an interesting study that aims to perform a global synthesis of the temporal and spatial patterns of the threats to rail bird species (Aves: Rallidae) while identifying the current conservation priorities and gaps. Although the data used in the study is freely available on the IUCN Red List of Threatened Species, we believe that it is very useful to analyse, concentrate and discuss this information in a single document. However, we have some issues regarding the methods, results, and discussion. Please find below comments on these issues, which we suggest that you take into consideration when reviewing the paper.
Response:	We thank you for your favorable assessment about the relevance of our study to the journal. We took on board your overall concerns and specific comments, focusing particularly on clarifying methods, results, and discussion (see response to comments below).
Comment 2:	We find that the methods are not always clear, especially due to the omission of important details of the data processing. We believe that, in its current version, the methods do not allow the readers to fully understand, repeat and validate the analyses.
Response:	We thank the referee for their comments' on how to improve the methods, and after addressing them (see response to comments below), we believe that the methods are now more readily understandable, with increased readability.
Comment 3:	In lines 115-117 you mention that you split category 5 in two sub-categories, 5.1 and 5.3. However, the script in the dryad repository ('Leveque_et_al._threats_global_and_island'), suggests that you used 5.1 and the sum of 5.2, 5.3, and 5.4, instead of 5.3. Is this correct? If so, we suggest that you clarify this in the text, since it will affect the interpretation of the results because the reader will not understand that those are related to three different sub-categories of threats (and not one).
Response:	We agree with the referee's comment and revised this section of the methods: "We split the threat '5. Biological resource use' in two categories: 'Hunting and collecting terrestrial animals' (5.1.) and 'Logging & indirect effects' (regrouping '5.2. Gathering terrestrial plants', '5.3 Logging and wood harvesting', and '5.4. Fishing & harvesting aquatic resources')" (lines 145-149).
Comment 4:	At the beginning of the methods, the names of the sub-categories of threat appear to be different from those used by the IUCN Red List; sub-category 5.1 is named 'Hunting & collecting terrestrial animals' and not 'Hunting and direct exploitation', and 5.3 is named 'Logging & wood harvesting' instead of 'Logging & indirect effects' (lines 105 and 106).
Response:	This echoes the sentiment of referee 1, that more consistency in the manuscript regarding the IUCN categories' names will help the reader's understanding of our analysis. Therefore, we renamed 5.1 for its original IUCN name 'Hunting & collecting terrestrial animals'. However, we kept 'Logging & indirect effects' (which is not the IUCN original name) as it regroups 3 subcategories of indirect effects '5.2. Gathering terrestrial plants', '5.3 Logging and wood harvesting', and '5.4. Fishing & harvesting aquatic resources' (lines 145-149).
Comment 5:	Likewise, in the conservation status' analysis, you write "We gathered the possible classifications in five categories: 'Research, Monitoring and Planning', 'Ecosystem Protection and Management', 'Species Management', 'Education & Awareness', and 'Law & Policy' (i.e., legislative protection)" (lines 134 to 137), but you do not define those categories.

Response:	By looking at the names of the categories in each scheme of the IUCN Red List, we can presume that: (i) 'Research, Monitoring and Planning' category was formed by grouping all categories in the 'Research Needed Classification Scheme'; (ii) 'Ecosystem Protection and Management' category was formed by grouping the first two categories, '1. Land/water protection' and '2. Land/water management', in the 'Conservation Actions Classification Scheme'; (iii) and the rest corresponded directly to each remaining category in the 'Conservation Actions Classification Scheme'. Furthermore, category 6 'Livelihood, economic & other incentives' of the 'Conservation Actions Classification Scheme' was not represented. The reason for this is not stated in the text, but one can assume that perhaps it did not appear in any of the studied species. These omissions impair the comprehension and reproducibility of the methods and, in our opinion, should be clarified. We welcome the referee's critique and re-wrote parts of the methods to better clarify and contextualize the use of the different categories (lines 175-185).
Comment 6: Response:	This part of the methods would probably be clearer if you add a diagram illustrating how the various parameters were put together to obtain the results. Thanks to the edits made for comments 3, 4, and 5, and another referee's comments, we restructured this paragraph, broken down by key analyses, and by adding sub-headings to the different parts of the methods and related results. We are now confident that thanks to this revision, the methods do not require an additional diagram.
Comment 7: Response:	There is a lack of information regarding the methods used to collect the data for extinct rail species. In the section 'Spatial and temporal patterns of threats', the only mention to the threats of extinct rail species is at the end: "Finally, we compared the contemporary threats with the historical causes of extinctions to assess temporal changes in the threat pattern and distribution of threatened and extinct species." (lines 124 to 126). In the dryad repository, there is an excel named 'Extinct_rails_database_2020' with information about the threats of each species, but there are almost no references and the methods used for data collection are not indicated. In our opinion, this information should be included in the methods. We accept the referee's point and re-wrote parts of the methods to make the information clearer that for extinct species too: threat data were taken from the IUCN website. (Lines 105-107 and 118).
Comment 8: Response:	Figure 1 could be improved by keeping the circles from covering the islands completely. In some cases, it becomes impossible to identify the island, weakening the usefulness of the map. Inserting the circles near the islands and connecting them with a line or arrow would avoid this problem, making the figure more informative. We appreciate the referee suggestion and edited the map to support it. We also updated the map and legend based on the other referee's suggestions.
Comment 9: Response:	You state that "Threats due to agriculture were mostly found in the Afrotropics and the Neotropics" (line 190). But Figure 2B indicates that agriculture was also considered a threat in 40% of the Palearctic rail species. It is not clear why you highlight the Neotropics, where only ~21-22% of the species were impacted by agriculture? There are different ways to read the figure: 1. Looking at the main threats impacting a bioregion (e.g., Palearctic species were mostly impacted by 'Natural system modifications' then 'Agriculture')

2. Looking at the figure focusing on each threat: in this case, we can see where the majority of species impacted are located (e.g., Agriculture: 15 species from Afrotropics and 11 species from Neotropics).
 We edited the sentence to make it clearer: “Threats due to agriculture mostly impacted species found in the Afrotropics and the Neotropics [...] (number of species, Fig. 2B)” (line 273).
 The figure was made this way so as to be consistent with Fig. 2A and allow for a clear comparison. We believe that the absolute number of rails impacted was a good measure of the impact a threat has. Alongside this, the percentages within a bioregion help the perception of what the main threats are.

Comment 10: **5. The discussion and conclusions of the manuscript lack the conservation focus that is promised in the goals.** *Currently, the discussion is largely a description of the results and is missing an actual debate on their implications to the conservation of rail bird species worldwide, or to the planning of specific conservation actions.*

For example, a discussion about why seven species recently improved their conservation status could provide clues on what kind of improvements should be made.

Response: We thank you for your comment and we addressed this idea in lines 428-436. We agree that adding more details on such information is indeed very useful for conservation strategies.
 Moreover, we fleshed out the overall discussion and conclusion further by answering comments 11, 12, and 13.

Comment 11: *Furthermore, in the abstract you indicate that “Forecasting the impacts of climate change on island endemic rails and disentangling the specific roles of extrinsic and intrinsic traits (like flightlessness), will be particularly valuable avenues of research for improving our forecasts of rail vulnerability.”. However, in the discussion, you do not elaborate on why climate change did not appear to be an important threat for rail species. A short discussion of this result would enrich this paragraph (lines 343 to 356).*

Response: We have revised this section of the discussion in accordance with these suggestions and included more examples and references. We believe that this edit greatly improves our argument (line 486-491).

Comment 12: *Likewise, concerning the countries identified both as rail ‘conservation hotspots’ and those most lacking in conservation efforts (i.e., Indonesia, the U.S.A., the United Kingdom, New Zealand and Cuba), you do not propose a justification for this result, or define specific actions to conserve rail species: “In that regard, we suggest that Indonesia, the U.S.A., the United Kingdom, New Zealand, and Cuba should be focusing more on measures to protect and recover their rail species, equally on their mainland and overseas territories.” (lines 336 to 338).*

Response: We thank the referee for suggesting this revision, as we agree that this will substantively improve the general conclusions and the implications for conservation. Therefore, we linked this section with our results from the conservation gaps identified per country (Table S4) and included a paragraph about it (lines 464-469, and lines 473-474).

Comment 13: *Finally, the discussion can be enriched by adding more references to support some of the explanations used to corroborate the results. For example, to justify why continental rails appear to be impacted by a wider diversity of threats, you merely say that this could be due to these species “being intrinsically more resistant to threats, even in synergy, or because fewer threats might naturally occur on islands” (lines 293 and 294). However, why would islands have fewer*

Response:	threats? Maybe you can explain this adding references and examples of threats that are less intense on islands. Once again, we welcome the referee's comment as it improved the quality of our arguments. We added more results about how threats impacted differently continental and island species (results section, lines 282-283). These, combined with the addition of more references in the discussion, allowed us to elaborate in a more precise and informative way regarding the origin of the discrepancy between continental and island species (discussion section, lines 394-400).
Comment 14:	Please write the word "family" in lowercase (line 21)
Response:	We accepted this suggestion (line 21).
Comment 15:	For the sake of consistency, we suggest that you always use either e.g. or e.g.,
Response:	We accepted this suggestion and used "e.g.," throughout the manuscript.
Comment 16:	Please add a final point after the parenthesis (line 57).
Response:	We accepted this suggestion (line 58).
Comment 17:	Improve coherence by changing the comma after "programs," to a semicolon (line 77), since it is the symbol used to separate the first and second goals.
Response:	We accepted this suggestion (line 78).
Comment 18:	To improve clarity, you may want to add 'so' to the following sentence: "Similarly, the New Caledonian rail (Gallirallus lafresnayanus) and the Samoan moorhen (Pareudiastes pacificus) are two 'Critically endangered' rail species that have not been seen with certainty since the 19th century and are suspected to be extinct (IUCN, 2019), so they were also excluded from our analysis (Table S2)" (lines 94 to 98).
Response:	We accepted this suggestion (line 100).
Comment 19:	Throughout the manuscript, five different spatial scales/levels are used across analyses: global (considering all continents and islands), continents (considering only continents), islands (considering only islands), bioregions (considering bioregions defined by Olson et al., 2001), and countries (used in the analyses focused on 'Conservation status and gaps'). The global scale is defined as 'worldwide' and 'globally', and the bioregions as 'bioregions' and 'bioregionally', which may make things less clear. We suggest that you use the same term to designate each level in the entire manuscript.
Response:	We accepted this suggestion and edit the manuscript to used 'globally' and 'bioregions' only.
Comment 20:	Please add the number of threats considered in the analysis of threat diversity (lines 102 to 107). If you used the first sub-category of threat as the level of threat and there are 45 sub-categories according to the IUCN Red List of Threatened Species, this means that the maximum number of threats for one species is 45, correct?
Response:	That is correct, the maximum number of threats is 45, however only 28 of these threats are found in rails. We added a sentence in the methods to incorporate this information (line 126-127).
Comment 21:	You should remove the comma after "severity," and add one after "timing" (line 110).
Response:	We accepted this suggestion (line 138).
Comment 22:	Indicate which version of the 'Threats Classification Scheme' was used in section 'Spatial and temporal patterns of threats'.
Response:	We used the IUCN - CMP Unified Classification of Direct Threats version 3.2, and added this information lines 136-137

Comment 23:	Please clarify that only two of the three levels considered for the timing of the threat were used in the analyses (past and ongoing). This information can be added, for example, in the following sentence “We included ‘Past’ impact to illustrate the temporal evolution of threats” (lines 112 and 113).
Response:	We revised this part of the methods to add “We included ‘Past’ and ‘Ongoing’ impact to illustrate the temporal evolution of threats (‘Future’ was not considered).” (lines 141-142).
Comment 24:	To use the same names throughout the manuscript (following the methods described in lines 103 and 104), we suggest that you rewrite the text “At this level, we used the first level of threat” to “For this analysis, we used the first category as the level of threat” (line 114).
Response:	We accepted this suggestion (line 144).
Comment 25:	To increase comprehension, we suggest that you change the text to “whether or not they were considered threatened by IUCN” (lines 119 and 120).
Response:	We accepted this suggestion (line 130).
Comment 26:	The “Research classification scheme” (line 130) corresponds to the ‘Research Needed Classification Scheme’? If yes, please replace it with its correct name and also include the accessed version.
Response:	We agree that this modification will make the methods clearer and added “Research needed classification scheme version 2.0” and the link to the framework, and repeated the edit for “Conservation actions classification scheme version 2.0” (lines 172-175).
Comment 27:	Remove “being” in “five were not being documented sufficiently to be confident about their flight ability” (lines 176 and 177).
Response:	We accepted this suggestion (line 245).
Comment 28:	We suggest that you include the total number of species (n = 140) considered in your database of threatened rail species, since it is the first time this database is mentioned in the results (line 178).
Response:	The total number of threatened rails in this study is 33 species. We edited the sentence to incorporate this information (line 249).
Comment 29:	In the legend of Table 1 please add an “s” to “scale” (...scales). Moreover, the table represents the number of threats and not the proportion of threats, right? We also suggest that you include the range of values considering all threatened rail species, that is the minimum and the maximum number of threats found.
Response:	We agree with the three points the referee raised and edited our manuscript to follow them.
Comment 30:	In Table 1, the median of the continental threatened rails is 4, and the first quartile is also 4. Please check if this is correct.
Response:	After careful verification, we confirm that this result is correct.
Comment 31:	Looking at your files ‘threat_type_database2019’ and ‘threats_region2019’, we assume that the sub-category 10 named ‘Geological events’ was not listed for any species because it was not listed in any of them. If this is the case, please add a sentence clarifying this issue in the results’ section ‘Spatial and temporal pattern of threats’.
Response:	Indeed, the sub-category ‘Geological events’ was not impacting any rail species, however, we added a line to clarify why (lines 149-150).
Comment 32:	To improve the clarity of Figure 2 and maintain the coherence among all figures we suggest that you: rewrite the description of panel (A) to match panel (B): “Threat diversity and intensity, measured as the impact score proposed by the IUCN Red List of Threatened Species, for rail species at a global scale (Gb; n =

	140), for island endemic rail species (Is; n = 42), and for continental rail species (Ct; n = 98)." (lines 207 and 208); change "per category" to "per bioregion" (line 210). Lastly, we suggest that the y-axis is renamed to "percentage of rail species impacted by threat".
Response:	We agree that these suggestions would improve the manuscript and readability, and edited the Figure 2 to incorporate them.
Comment 33:	Still concerning Figure 2, the first panel (A) also does not include the categories "Other" and "Energy production and mining". Please explain this as you did for panel (B).
Response:	We edited the Figure 2B legend to match the one of Figure 2A.
Comment 34:	Remove "proportionally" in the following sentence: "'Pollution' and 'Natural system modifications' (e.g., fire, dams, water abstraction) were the only threats that were proportionally more threatening on continents than for island endemics (Fig. 2A)." to "'Pollution' and 'Natural system modifications' (e.g., fire, dams, water abstraction) were the only threats that were more threatening for continental rails than for island endemics (Fig. 2A)." (lines 194 to 196).
Response:	We accepted this suggestion (line 288).
Comment 35:	We suggest that you change the location of the word "globally" and specify the panel of Figure 2: "Globally, less than 20% of rails are impacted by 'Natural System modifications', however, it is an important factor in the Palearctic, where it impacts 60% of that region's rails (Fig. 2B)." (lines 196 to 198).
Response:	We accepted this suggestion (lines 283-284).
Comment 36:	It is unclear why the category "Research, Monitoring and Planning" was named "Research & monitoring" in line 212. Are you considering the categories 'Research' and 'Monitoring'? If yes, why would you say "We gathered the possible classifications in five categories" (lines 134 and 135)? Please clarify this issue to avoid confusion.
Response:	We welcome the referee's comment and clarified this category name by naming it 'Research & Monitoring' from the point where we first refer to it (line 180) and throughout the manuscript. We described lines 182-184 what sub-categories are included to increase understanding of the methods.
Comment 37:	To improve clarity, we suggest that you rewrite the following sentence "including six species that became newly threatened" to "including six species that were again classified as threatened" (lines 226 and 227), and add "Out of those 11 species," before "Eight species were endemic...", and remove "species" after "Eight..." (line 227).
Response:	We accept that our original writing could be confusing and therefore we rephrased this sentence to "including six species that were allocated a threatened status for the first time" as we believe this explains it more precisely. We edited the other sentences to include the referee's modifications (lines 331-332).
Comment 38:	Add a reference to this statement: "Most of the past rail extinctions occurred on the Pacific islands, directly linked to the fact that they were the support for the largest radiation of rails, reaching high levels of endemism." (lines 258 to 260).
Response:	We added four references to support this statement (lines 365-366).
Comment 39:	We suggest that you remove "in general" (line 261) and reword the sentence to: "Flightlessness is an evolutionary trait that has been found to make bird species more extinction-prone during different waves of extinction (Duncan et al., 2002; Boyer, 2008), especially rails (Steadman, 1995; Curnutt & Pimm, 2001), and extinction-prone during different waves of extinction s an important contributor to the vulnerability of rails to contemporary threatening processes."

Response:	We edited this sentence with the referee's suggestions and in a way to make the sentence clearer to read (lines 367-369).
Comment 40:	You may want to replace "globally" with 'worldwide' to avoid using the same word many times (line 270).
Response:	We agree with this comment and edited the text (line 376-377).
Comment 41:	Provide references for the following affirmation "However, Oceania, which in ancient times supported hundreds of flightless rails, now stands bleakly as a rail-species graveyard after the Holocene mass extinction (only five rail species survive in the Pacific basin, including just two endemics to the region)." (lines 285 to 288).
Response:	We added two references to support this statement (line 393).
Comment 42:	We find the font used in Figure 2 a bit too small to be easily readable.
Response:	We agree with this comment and edited the Figure so we could increase the font size.